# Impact of Climate Change on the Water Requirements of Oat in Northeast and North China

**Hao Jia** [1,2] **, Ting Zhang** [1,2]**, Xiaogang Yin** [1,2]**, Mengfei Shang** [1,2]**, Fu Chen** [1,2]**, Yongdeng Lei** [1,2,]*****
**and Qingquan Chu** [1,2,]*****

[1]    College of Agronomy and Biotechnology, China Agricultural University, Beijing 100193, China;
       jiahao@cau.edu.cn (H.J.); zhangting0426@cau.edu.cn (T.Z.); yinxg@cau.edu.cn (X.Y.);
       caushang@cau.edu.cn (M.S.); chenfu@cau.edu.cn (F.C.)
[2]    Key Laboratory of Farming System, Ministry of Agriculture and Rural Affairs, Beijing 100193, China
*****    Correspondence: cauchu@cau.edu.cn (Q.C.); leiyd@cau.edu.cn (Y.L.); Tel.: +86-10-62731207

**Abstract:** Crop water requirements are directly affected by climatic variability, especially for crops grown in the areas which are sensitive to climatic change. Based on the SIMETAW model and a long-term meteorological dataset, we evaluated the spatiotemporal variations of climatic change impacts on water requirement of oat in North and Northeast China. The results indicated that effective rainfall showed an increasing trend, while the crop water requirement and irrigation demand presented decreasing trends over the past decades. The water requirement of oat showed significant longitudinal and latitudinal spatial variations, with a downtrend from north to south and uptrend from east to west. Climatic factors have obviously changed in the growth season of oat, with upward trends in the average temperature and precipitation, and downward trends in the average wind speed, sunshine hours, relative humidity, and solar radiation. Declines in solar radiation and wind speed, accompanied with the increase in effective rainfall, have contributed to the reduced crop water requirement over these decades. Given the complex dynamic of climate change, when studying the impact of climate change on crop water requirements, we should not only consider single factors such as temperature or rainfall, we need to analyze the comprehensive effects of various climatic factors.

**Keywords:** climate change; crop water requirement; temporal and spatial variations; oat; Northeast and North China

## 1. Introduction

The impact of climate change on water resources and crop production is a major problem which should be given attention by China and the rest of the world in the twenty-first century [1], especially, in Northeast and North China, which have large areas of cultivated lands and irrigation areas, but have less than 20% of the total water resources [2]. In the past few decades, the decreased precipitation associated with increasing temperature made frequent the occurrence of drought, which is the cause of substantial yield loss in China [3–5]. The significant upward trend in high temperature extremes accompanied with variable rainfall [6] could cause frequent droughts [7] and water shortages [8]. Due to the highly unmatched agricultural water and soil resources, irrigation water has become a restrictive condition to crop production and the sustainable development of agriculture in Northeast and North China [9–12]. It was projected that the drought will be more serious with the increasing temperature [13]. Therefore, identifying how to improve the utilization of natural rainfall in the arid and semi-arid areas of northern China is essential to mitigate water stress and to ensure regional food security.

The crop water requirement is the sum of water consumption, which includes crop transpiration and soil evaporation, during the growing period. In recent decades, several models based on Penman–Monteith equation [14] have been employed to calculate crop water requirements, such as the CROPWAT model [15] and SIMETAW (Simulation Evapotranspiration of Applied Water) model [16,17]. As the crop water requirement is directly affected by climatic variability, increasing studies have focused on the impacts of climate change on crop water requirements. Some effective model tools, including the MAGICC/SCENGEN 5.3 compound model [18], have been used to evaluate the impacts of climate change on the crop water requirements in various regions. Generally, climatic variability has inevitable impacts on crop production and could shorten the crop growing period and affect water demand. Therefore, some cultivation management measures are required to compensate for the negative impact of climate change on crop growth [19,20].

In terms of crop varieties, current studies on crop water requirements have mainly concentrated on major crops, such as paddy rice, wheat, and maize [21–23]. By contrast, little attention has been given to certain minor—but very important—cereals, such as oat (*Avena sativa* L.). Oat is mainly cultivated in the north, northeast, and southwest alpine regions of China [24,25]. Various studies have confirmed that an increase in annual temperature and irregular changes in precipitation might influence the crop water requirement [26]; however, the mechanism of the influencing process is still unclear. Therefore, this study aims to analyze the temporal and spatial variation in the water requirement (*ETc*), effective rainfall (*Er*), and irrigation demand (*ETaw*) of oat during its growing season in North and Northeast China. In addition, the tendency of climatic factors was analyzed by explaining the correlation between crop water requirements and climate change. The expected results could be used to address the impact of climate change on the water requirement of oat, and this information could be used to develop effective strategies to mitigate the adverse effects of climate change.

## 2. Materials and Methods

### 2.1. Study Area

We selected 15 meteorological stations to analyze the water requirement of oat in North and Northeast China based on the experimental records. These stations included 7 stations in Inner Mongolia Autonomous Region, 2 stations in Heilongjiang Province, 1 station in Jilin Province, 1 station in Liaoning Province, 2 stations in Hebei Province, and 2 stations in Shanxi Province (Figure 1). In these areas, oat is one of the most important food sources for local residents.

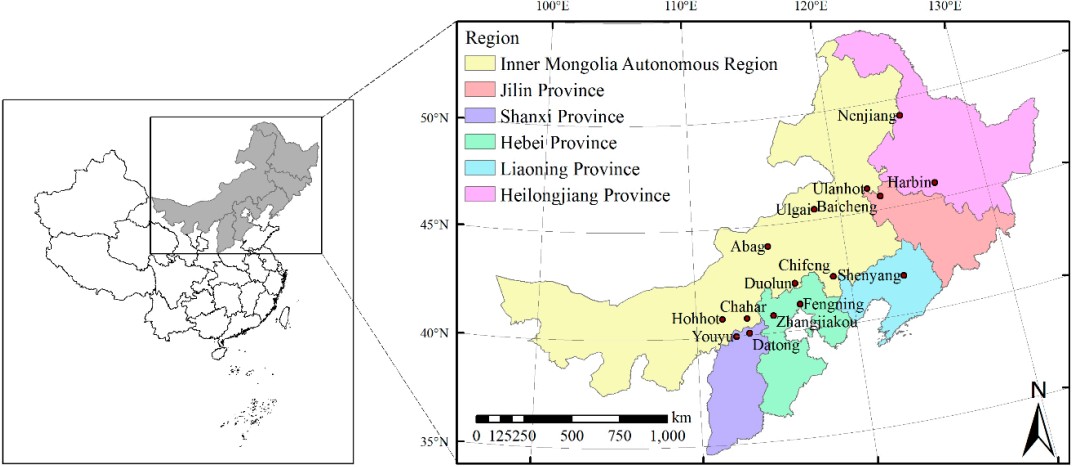

**Figure 1.** Main planting areas of oat in North and Northeast China.

## 2.2. Data Sources

From 2010 to 2015, we conducted field experiments on oat in our study areas, and we collected empirical data related to crop growth, including sowing and harvest dates, cultivated area, planting density, and fertilizer management. Soil data needed for models were from the National Soil Survey Databases, which included soil types, soil texture, wilting coefficient, soil bulk density, saturated moisture content, etc. The climatic data were derived from the standard meteorological stations in China during the period of 1960–2015, and these data included daily minimum temperature, mean temperature, maximum temperature, precipitation, wind speed, sunshine duration, and relative humidity. These types of data are all useful for calculating the water requirement and irrigation demand of oat.

## 2.3. Calculation of Water Requirement, Effective Rainfall, and Irrigation Demand

Many models can be used to evaluate the crop water requirements, including APSIM [27] and AquaCrop [28], etc. However, these models are more focused on simulating crop yield under different moisture conditions. In this study, the SIMETAW model was adopted to analyze the water requirements of oat during the growth period. The SIMETAW model is a widely used soil water balance model with the advantages of friendly interface, good interpretability, and reliable results [16,17,29]. Besides, the SIMETAW model is compatible with multiple types of climatic data, and its required parameters are relatively easier to obtain compared to other models [30]. In recent years, this model has been widely used in evaluating the water requirements of maize, winter wheat, spring wheat, and soybean [16,17].

In this study, the water requirement (*ETc*), effective rainfall (*Er*), and irrigation demand (*ETaw*) of oat were simulated by the SIMETAW model. The *ETc* was calculated using Equation (1):

$$ETc = Kc \times ETo,\tag{1}$$

where *Kc* is the crop coefficient, and *ETo* represents the reference crop evapotranspiration, which is calculated from various climatic data, including daily temperature, rainfall, sunshine duration, and relative humidity. *Kc* values of oat are stored in the SIMETAW model, which are recommended by FAO-56 [31,32] and are calibrated based on the climatic data, crop, and soil parameters in different regions. *ETo* is calculated with a modified version of the Penman–Monteith equation [29,32,33]:

$$ETo = \frac{0.408\Delta(R_n - G) + \gamma \frac{900}{T+273} u_2(e_s - e_a)}{\Delta + \gamma(1 + 0.34u_2)},\tag{2}$$

where $\Delta$ is the slope of the saturation vapor pressure at mean air temperature curve (kPa $°\text{C}^{-1}$), $R_n$ and $G$ are the net radiation and soil heat flux density in MJ $\text{m}^{-2}$ $\text{d}^{-1}$, $\gamma$ is the psychrometric constant (kPa $°\text{C}^{-1}$), $T$ is the daily mean temperature (°C), $u_2$ is the mean wind speed in m $\text{s}^{-1}$, $e_s$ is the saturation vapor pressure (kPa) calculated from the mean air temperature (°C) for the day, and $e_a$ is the actual vapor pressure (kPa) calculated from the mean dew point temperature (°C) for the day. The effective rainfall (*Er*) can be calculated by Equations (3) and (4):

$$Er = \sum_{i=1}^{n} Er_i,\tag{3}$$

$$Er_i = \begin{cases} Pcp, Pcp < CETc \\ CETc, Pcp \geq CETc \end{cases},\tag{4}$$

where *Er* is the effective rainfall during the whole growth period of oat (mm), $Er_i$ is the daily effective rainfall (mm), *n* is the number of days within the growth period of oat, *Pcp* is the actual daily rainfall (mm), and *CETc* is the daily water requirement (*ETc*) of oat (mm), which is calculated using the daily data of *Kc* and *ETo*.

The irrigation demand (*ETaw*) of oat was calculated using Equation (5):

$$ETaw = ETc - Er. \tag{5}$$

In the process of model simulation, it is necessary to input the climatic data, the soil and crop parameters mentioned above.

### 2.4. Spatial Analysis and Statistical Methods

The spatial distributions of the oat water requirements were analyzed based on the inverse distance weighted (IDW) interpolation method using ARCGIS 10.0. IDW is one of the most practical interpolation methods for displaying and mapping the spatial distributions of climatic variations [34]. Besides, we adopted the SPSS statistics model to further analyze the correlations between oat water requirements and climatic factors. The specific statistical methods included regression analysis, correlation analysis, and variance analysis, and we conducted Duncan's multiple-range test at the 0.05 probability level.

## 3. Results

### 3.1. The Temporal Variation in Oat Water Requirements in North and Northeast China from 1960 to 2015

In the past 56 years, the mean annual *ETc* and *ETaw* of oat showed decreasing tendencies, while the *Er* showed an increasing trend in North and Northeast China. The descending rate of *ETc* was greater in North China than in Northeast China, which decreased by 2.3 and 1.7 mm per decade, respectively, with a range of 301.0 to 367.8 mm in North China and 299.8 to 381.5 mm in Northeast China, respectively. The mean values of *ETc* were found to be slightly higher in Northeast China than in North China, and the maximum and minimum values were 344.4 mm and 340.1 mm in the 1960s, respectively, and 330.3 mm and 328.5 mm in the 1990s, respectively (Table 1).

**Table 1.** Decadal variation in oat water requirements in North and Northeast China (mm).

| Decades | North China | | | Northeast China | | |
| --- | --- | --- | --- | --- | --- | --- |
| | *ETc* Water Requirement | *Er* Effective Rainfall | *ETaw* Irrigation Demand | *ETc* Water Requirement | *Er* Effective Rainfall | *ETaw* Irrigation Demand |
| 1960–1969 | 340.1 | 108.4 | 231.7 | 344.4 | 148.8 | 195.6 |
| 1970–1979 | 335.7 | 110.5 | 225.2 | 340.5 | 161.1 | 179.3 |
| 1980–1989 | 339.1 | 106.8 | 232.3 | 343.5 | 161.9 | 181.7 |
| 1990–1999 | 328.5 | 123.8 | 204.6 | 330.3 | 175.3 | 154.9 |
| 2000–2015 | 320.1 | 106.0 | 214.1 | 338.6 | 160.0 | 178.8 |

The *Er* of oat means that part of the rainfall is utilized directly or indirectly to satisfy crop evapotranspiration and field water consumption during the growth period. Moreover, the *Er* of oat ranged between 73.1 and 192.3 mm in North China, and between 98.7 and 220.4 mm in Northeast China. Additionally, the value statistically significantly decreased from 123.8 mm in the 1990s to 106.0 mm in the 2000s. The mean annual *Er* of oat was lower in North China than that in Northeast China, while the values of *ETaw* presented an opposite trend (Table 1). The *ETaw* of oat, which supplements insufficient natural rainfall, was calculated as having insignificant decreasing trends in both North and Northeast China. However, the *ETaw* of oat in the 1990s (204.6 mm) was significantly lower than that of the 1980s (232.3 mm) and that of the 2000s (214.1 mm) in North China (Table 1).

The mean values of *ETc*, *Er*, and *ETaw* varied considerably across the different regions, with ranges greater than 50 mm and 100 mm, respectively (Figure 2). The *ETc* of oat greatly significantly decreased ($p < 0.01$) in Youyu County, Zhangjiakou City, and Chahar Right Front Banner; additionally, the *ETc* of oat significantly decreased ($p < 0.05$) in Baicheng and Shenyang city. Furthermore, it displayed

increasing trends in Hohhot City, Datong City, and Nenjiang County, and it significantly increased ($p < 0.05$) in Nenjiang County. By contrast, the tendency of *Er* showed an insignificant overall increase among stations, with the exceptions of Fengning County and Abag Banner. However, the annual *ETaw* of oat reflected a tendency to decrease across regions apart from Hohhot City, Datong City, and Fengning and Nenjiang counties, and it significantly declined ($p < 0.05$) in Zhangjiakou City.

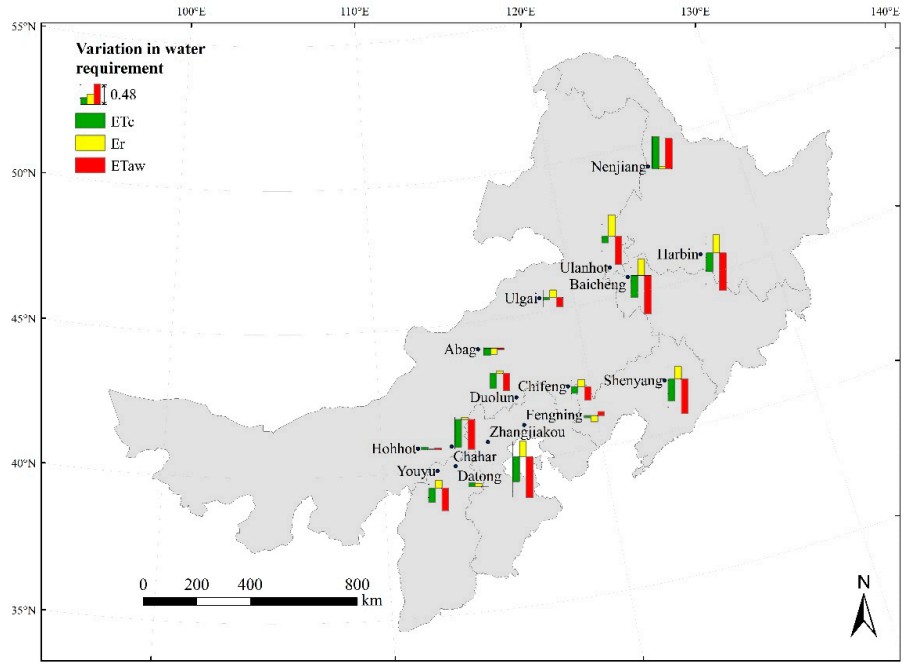

**Figure 2.** Variation in oat water requirements at 15 stations.

### 3.2. The Spatial Variation in Oat Water Requirements in North and Northeast China from 1960 to 2015

The mean annual *ETc*, *Er*, and *ETaw* varied significantly across the regions of North and Northeast China (Table 2). The characteristic distribution of *ETc* was recorded to be higher in the northern and western areas than in the southern and eastern areas (Figure 3a). The mean value of *ETc* was larger than 338.6 mm in Northeast China, and larger than 334.9 mm in North China. The maximum and minimum values of *ETc* were calculated as 362.2 mm in Baicheng City and 305.7 mm in Zhangjiakou City, respectively, with a total range of 56.5 mm.

**Table 2.** Spatial variation in oat water requirements at 15 stations (mm).

| Station | *ETc* Water Requirement | *Er* Effective Rainfall | *ETaw* Irrigation Demand |
|---|---|---|---|
| Duolun County | 357.8 | 148.8 | 209.0 |
| Fengning County | 317.6 | 156.4 | 161.2 |
| Youyu County | 316.9 | 125.7 | 191.2 |
| Zhangjiakou City | 305.7 | 88.8 | 216.9 |
| Hohhot City | 332.6 | 105.9 | 226.7 |
| Chahar Right Front Banner | 345.2 | 88.0 | 257.1 |
| Datong City | 351.3 | 113.6 | 237.7 |
| Ulgai District | 336.8 | 88.9 | 247.9 |
| Abag Banner | 349.9 | 84.2 | 265.6 |
| Baicheng City | 362.2 | 132.2 | 230.0 |
| Chifeng City | 341.9 | 115.3 | 226.6 |
| Harbin City | 339.3 | 173.5 | 165.8 |
| Nenjiang County | 355.1 | 225.9 | 129.2 |
| Shenyang City | 331.2 | 182.6 | 148.6 |
| Ulanhot City | 308.4 | 137.3 | 171.1 |

The *Er* of oat was calculated to be higher in the northern and eastern areas than in the southern and western areas of China, with maximum and minimum values of 225.9 mm in Nenjiang County and 84.2 mm in Abag Banner, respectively, with a range of 141.7 mm (Figure 3b). Differences in the *Er* across the region were mostly attributed to the regional variation in natural rainfall and the efficient utilization of rainfall during the oat growing season; this conclusion was based on the cultivation condition of oat, the soil types, the surface runoff, and the deep leakage.

The average minimum and maximum values of *ETaw* were 129.2 mm in Nenjiang County and 265.6 mm in Abag Banner, respectively, with the largest range of 136.4 mm (Figure 3c). In addition, the variation in the *ETaw* was closely related to the variability of the *ETc* and *Er*.

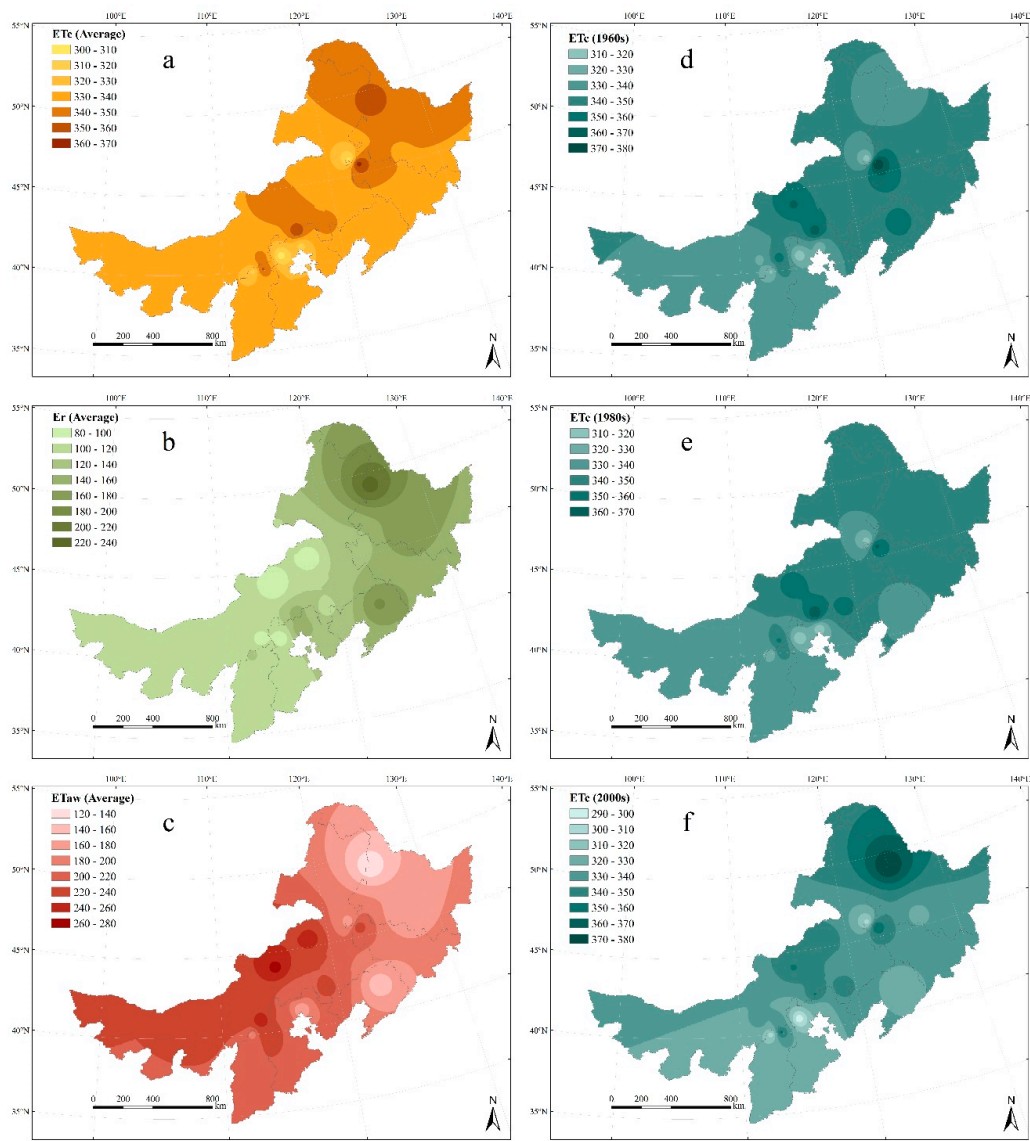

**Figure 3.** Spatial variation in oat water requirements (mm) ((**a**): water requirement; (**b**): effective rainfall; (**c**): irrigation demand) during the past five decades ((**d**): the 1960s; (**e**): the 1980s; (**f**): the 2000s) in North and Northeast China.

The mean decadal *ETc* obviously varied across the region between 1960 and 2015, and the calculated values were higher in eastern Heilongjiang, northern Inner Mongolia, and Jilin Province; by contrast, overall, the values were lower in northeast Inner Mongolia and northwest Hebei Province (Figure 3d–f). During the 1960s, the maximum *ETc* was found in northern Jilin Province, while the

minimum values occurred in northeast Inner Mongolia, northwest Hebei Province, and northern Shanxi Province. The lower values were found in northeast Inner Mongolia and northern Shanxi Province. However, *ETc* marginally increased in the 1980s. The mean value was higher in northern Jilin and Hebei provinces than that of northeast Inner Mongolia and those of northeast and northern Hebei Province. In addition, the *ETc* of oat still appeared higher in northern Inner Mongolia and eastern Heilongjiang Province, and it was lower in northwest Hebei Province from 2000 to 2015.

### 3.3. The Variation in Climatic Factors during the Oat Growing Season from 1960 to 2015

At the 15 climatic stations, the mean temperature and precipitation of most sites had increasing tendencies from 1960 to 2015, while there were declining trends in mean wind speed, sunshine duration, mean relative humidity, and solar radiation (Table 3). The mean decadal temperature increased 0.11–0.43 °C during the oat growth period, with a highly statistically significant ($p < 0.01$) upward trend. The maximum increase rate for temperature was found in Hohhot City, and the increase rate of 0.3–0.4 °C per decade was found in Datong City, Ulgai District, Abag Banner, Harbin City, Ulanhot City, and Nenjiang County. By contrast, the mean wind speed, sunshine duration, and solar radiation of most sites indicated overall downtrends ($p < 0.05$), with ranges of −0.025 to −0.553 m/s, −0.034 to −0.342 h/day, and −0.047 to −0.468 MJ/m$^2$ per decade, respectively. However, compared to the other four climatic factors, the changes in mean precipitation and relative humidity of most sites were relatively not significant.

**Table 3.** Variations in climatic factors during the oat growth period at 15 stations from 1960 to 2015 (annual average/slope value).

| Station | Mean Temperature (°C) | Daily Precipitation (mm) | Mean Wind Speed (m/s) | Sunshine Duration (hours/day) | Mean Relative Humidity (%) | Solar Radiation (MJ/m$^2$) |
|---|---|---|---|---|---|---|
| Duolun County | 13.09/0.0285 ** | 1.62/−0.0036 | 3.62/−0.0146 ** | 9.17/−0.0331 ** | 53.87/−0.0261 | 22.31/−0.0452 ** |
| Fengning County | 17.80/0.0130 | 2.06/0.0013 | 2.67/−0.0182 ** | 8.53/−0.0114 * | 53.21/0.0165 | 21.77/−0.0159 * |
| Youyu County | 15.03/0.0228 ** | 1.58/0.0017 | 2.67/−0.0182 ** | 9.14/−0.0121 | 51.98/−0.0983 * | 22.76/−0.0171 * |
| Zhangjiakou City | 17.21/0.0280 ** | 1.21/0.0077 * | 2.79/−0.0306 ** | 8.87/−0.0205 ** | 41.57/−0.0165 | 21.93/−0.0284 ** |
| Hohhot City | 17.80/0.0430 ** | 1.31/0.0001 | 2.18/0.0011 | 9.24/−0.0185 ** | 43.83/−0.1977 ** | 22.80/−0.0258 ** |
| Chahar Right Front Banner | 14.42/0.0282 ** | 1.16/0.0020 | 4.10/−0.0553 ** | 9.73/0.0058 | 42.65/−0.0689 | 23.39/0.0079 |
| Datong City | 17.50/0.0317 ** | 1.38/0.0005 | 3.15/−0.0025 | 8.65/−0.0113 * | 46.23/−0.1171 ** | 22.06/−0.0160 * |
| Ulgai District | 14.59/0.0357 ** | 1.09/0.0011 | 3.74/−0.0273 ** | 9.43/−0.0062 | 45.58/−0.0359 | 22.44/−0.0082 |
| Abag Banner | 14.51/0.0375 ** | 1.05/−0.0024 | 4.15/−0.0223 ** | 9.62/−0.0206 ** | 43.16/−0.0795 | 22.90/−0.0277 ** |
| Baicheng City | 17.04/0.0290 ** | 1.73/−0.0014 | 4.00/−0.0301 ** | 8.77/−0.0262 ** | 51.44/0.0015 | 21.40/−0.0352 ** |
| Chifeng City | 17.54/0.0113 | 1.44/0.0036 | 2.79/−0.0096 * | 8.91/−0.0034 | 45.34/0.0544 | 22.03/−0.0047 |
| Harbin City | 16.39/0.0389 ** | 2.05/0.0090 | 3.82/−0.0502 ** | 8.17/−0.0229 ** | 57.53/−0.0134 | 20.61/−0.0303 ** |
| Nenjiang County | 17.04/0.0377 ** | 2.63/0.0025 | 3.72/0.0046 | 8.73/0.0031 | 63.99/−0.1297 ** | 21.12/0.0024 |
| Shenyang City | 17.88/0.0168 * | 2.33/0.0092 | 3.39/−0.0178 ** | 7.90/−0.0342 ** | 59.78/0.0218 | 20.65/−0.0468 ** |
| Ulanhot City | 18.04/0.0365 ** | 2.12/0.0071 | 3.31/−0.0269 ** | 8.98/−0.0153 ** | 50.22/−0.0582 | 21.96/−0.0205 ** |

Note: Slope values represent the tendency of historical changes in climate factors. A positive value indicates an upward trend, and a negative value shows a downtrend trend. * and ** indicate significant variations at the 0.05 and 0.01 levels, respectively.

### 3.4. Impact of Climate Change on Oat Water Requirements during the Growth Period

Positive correlations were found between the oat water requirement (Table 2) and the solar radiation, mean temperature, wind speed, and sunshine duration. A negative correlation appeared between the oat water requirement and the daily precipitation and mean relative humidity (Table 4). Solar radiation, mean wind speed, sunshine duration, and mean relative humidity were predicted to have dominant impacts on the water requirements of oat, with possible strong correlations with the oat water requirement of 0.229–0.674, 0.181–0.697, 0.304–0.716, and 0.186–0.700, respectively; these values were highly statistically significant ($p < 0.01$) at more than 12 stations. However, the impacts of the climatic factors on the water requirements varied across the region during the oat growth period, which may have contributed to the regional variation in the water requirements.

**Table 4.** Correlation of oat water requirement with climatic factors in North and Northeast China from 1960 to 2015.

| Station | Mean Temperature (°C) | Daily Precipitation (mm) | Mean Wind Speed (m/s) | Sunshine Duration (hours/day) | Mean Relative Humidity (%) | Solar Radiation (MJ/m$^2$) |
|---|---|---|---|---|---|---|
| Duolun County | 0.137 | −0.250 | 0.434 ** | 0.526 ** | −0.541 ** | 0.531 ** |
| Fengning County | 0.361 ** | −0.220 | 0.594 ** | 0.318 * | −0.389 ** | 0.324 * |
| Youyu County | 0.054 | −0.133 | 0.509 ** | 0.537 ** | −0.387 ** | 0.542 ** |
| Zhangjiakou City | 0.020 | −0.433 ** | 0.598 ** | 0.449 ** | −0.430 ** | 0.453 ** |
| Hohhot City | 0.215 | −0.278 * | 0.605 ** | 0.407 ** | −0.441 ** | 0.410 ** |
| Chahar Right Front Banner | 0.087 | −0.225 | 0.596 ** | 0.181 | −0.304 * | 0.186 |
| Datong City | 0.388 ** | −0.106 | 0.511 ** | 0.369 ** | −0.441 ** | 0.371 ** |
| Ulgai District | 0.327 * | −0.502 ** | 0.391 ** | 0.434 ** | −0.716 ** | 0.438 ** |
| Abag Banner | 0.268 | −0.385 ** | 0.360 ** | 0.534 ** | −0.615 ** | 0.538 ** |
| Baicheng City | 0.151 | −0.256 | 0.462 ** | 0.501 ** | −0.579 ** | 0.505 ** |
| Chifeng City | 0.421 ** | −0.307 * | 0.559 ** | 0.318 * | −0.558 ** | 0.325 * |
| Harbin City | 0.163 | −0.343 * | 0.424 ** | 0.697 ** | −0.402 ** | 0.700 ** |
| Nenjiang County | 0.625 ** | −0.348 * | 0.229 | 0.663 ** | −0.547 ** | 0.652 ** |
| Shenyang City | 0.304 * | −0.253 | 0.674 ** | 0.539 ** | −0.631 ** | 0.541 ** |
| Ulanhot City | 0.285 * | −0.539 ** | 0.342 * | 0.423 ** | −0.577 ** | 0.425 ** |

Note: * and ** indicate significant correlations at the 0.05 and 0.01 levels, respectively.

In the last 56 years, the water requirement of oat was predicted to increase because the mean temperature increased significantly, and the mean relative humidity decreased. However, the water requirement of oat showed an overall decreasing trend in North and Northeast China when the solar radiation, mean wind speed, and sunshine duration declined, and the daily precipitation increased. The variations in the solar radiation, mean wind speed, and sunshine duration played leading roles in the water requirements of oat.

However, the impacts of climate change on oat water requirements had a marginal regional difference. At some stations, the water requirement of oat increased, which could be related to the landform as well as the altitude of certain areas. In addition, the variability of the dominant climatic factors, such as the mean temperature and relative humidity, resulted in a significant increasing trend in oat water requirement at Nenjiang station. The correlations between the water requirement and the mean temperature and relative humidity were highly statistically significant at the 0.01 level, with correlation coefficients of 0.625 and 0.547, respectively. The effect of mean wind speed on the water requirement of oat was not significant.

## 4. Discussion

The impact of climate change on crop water requirement is very complex due to the variability of climatic factors; as a result, there are only hypothesized explanations about the variation trend of water requirements [35]. Though temperature had a significant uptrend in the period of 1960–2015, other climatic factors, such as sunshine duration, mean wind speed, and solar radiation, showed different variations over the past 56 years. The interactive influences of different climatic factors on the water requirement of oat varied across different regions. The decrease in mean wind speed and solar radiation would lead to an expected decrease of water requirements, while an increase of mean temperature and decrease of relative humidity led to the opposite, but failed to offset the impacts of wind speed and solar radiation on water requirements. Therefore, there was a declining trend for the oat water requirement.

Although more research has been carried out on the impacts of climate change on the water requirements of some major crops, including rice, maize, and wheat, studies on the impact of climate change on the water requirements of oat have rarely been reported. The results of our research have some similarities to related studies on other crops. According to Yoo et al. [36], in future decades, the temperature in parts of South Korea will increase, but the water demand for rice will decrease, which will be mainly caused by the decline in solar radiation. Liu et al. [21,22] reported that in North China, water requirements of winter wheat and summer maize in most locations showed a

downtrend in the past 50 years, which was mainly attributed to the reduction of sunlight hours and wind speed. According to Yang et al. [16,17,23], in the Huang-Huai-Hai farming region of China, the water requirements of summer maize and spring maize, as well as winter wheat, showed obvious downtrends in the past half century, which were primarily attributed to the reduction of solar radiation.

Given the complex dynamic of climate change, when studying the impact of climate change on crop water requirements, we should not only consider single factors, such as temperature or rainfall—we need to analyze the comprehensive effects of various climatic factors. Besides, the impact of climatic factors on the water requirement of oat varied among different regions, which may be connected in terms of local soil types, geographic and geomorphologic conditions, and altitude. Moreover, it is necessary to investigate the effects of field microclimate, regional water consumption structure, crop variety, sowing date, and cultivation management on crop water requirements.

## 5. Conclusions

In this study, we used the SIMETAW model to evaluate the temporal and spatial variations in oat water requirements, as well investigated the impacts of climatic variability on the oat water requirements in North and Northeast China during the last half century. From 1960 to 2015, the *ETc* and *ETaw* of oat decreased, while the *Er* showed an increasing trend. The mean *ETc* ranged between 301.0–367.8 mm and 299.8–381.5 mm in North and Northeast China, respectively. Statistical differences in oat water requirements were found among various stations, and the largest spatial differences were greater than 50, 100, and 100 mm for *ETc*, *Er*, and *ETaw,* respectively. The spatial variation in *ETc* was higher in eastern Heilongjiang, northern Inner Mongolia and Jilin provinces, while it was lower in northeast Inner Mongolia and northwest Hebei Province.

In terms of climatic factors, the mean temperature of most stations showed a statistically significant increasing trend, while the wind speed, sunshine duration, and solar radiation decreased significantly. There were no significant trends for the mean daily precipitation ($p > 0.05$) and the relative humidity ($p > 0.05$). The mean temperature, wind speed, sunshine duration, and solar radiation were positively correlated with the *ETc*, while the daily precipitation and relative humidity were negatively correlated with the *ETc*. In addition to the increase in effective rainfall, the negative trend of the oat water requirement was mainly attributed to the declining solar radiation and wind speed. In view of the variability of different climatic factors, such as a significantly increased mean temperature during the growth season of oat, we need to take necessary strategies to adapt to the impacts of climate change on oat water requirements. For example, the impacts of climate change on water requirements of oat could be alleviated through properly adjusting the sowing date, in order to match it with the changing rhythm of climatic factors during the growth season.

**Author Contributions:** Conceptualization, Y.L. and Q.C.; Formal analysis, H.J., F.C. and Q.C.; Funding acquisition, Q.C.; Investigation, H.J. and M.S.; Project administration, F.C. and Q.C.; Software, T.Z.; Supervision, Y.L. and Q.C.; Visualization, H.J.; Writing—original draft, H.J.; Writing—review & editing, T.Z., X.Y. and Y.L.

**Funding:** This research was funded by the National Natural Science Foundation of China (31871581; 31801315), the National Key Research and Development Program of China (2016YFD0300201), and the National Buckwheat Industry Technology System (CARS-08).

**Acknowledgments:** The authors would like to thank the anonymous reviewers for their helpful comments and suggestions.

**Conflicts of Interest:** The authors declare no conflict of interest.

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
