# Peer review of "Impact of Climate Change on the Water Requirements of Oat in Northeast and North China"

_water, doi:10.3390/w11010091_

Round 1

Reviewer 1 Report

The paper aims at analyzing the impact of climate change on the Oat water requirement t in Northeast and North China.

Include some key-data in the abstract.

The authors also present experimental results on real sensor data to show that the conclusion is effective. Authors used a SIMETAW model to estimate water requirement, effective rainfall and irrigation demand of Oat. The work seems to describe the spatiotemporal variations of climatic change using the climate data in 15 meteorological stations.

However, appears weak and unclear in model presentation and discussion.

The data in Table 3 and Table 4 are the average values of the climate variables in almost 50 years, cannot support the argument about the impact of temporal variations of climate change on Oat water requirement.

Other minor comments are shown in the manuscript.

Author Response

Point 1:

Comments and Suggestions for Authors

The paper aims at analyzing the impact of climate change on the Oat water requirement in Northeast and North China.

Include some key-data in the abstract.

The authors also present experimental results on real sensor data to show that the conclusion is effective. Authors used a SIMETAW model to estimate water requirement, effective rainfall and irrigation demand of Oat. The work seems to describe the spatiotemporal variations of climatic change using the climate data in 15 meteorological stations.

However, appears weak and unclear in model presentation and discussion.

The data in Table 3 and Table 4 are the average values of the climate variables in almost 50 years, cannot support the argument about the impact of temporal variations of climate change on Oat water requirement.

Other minor comments are shown in the manuscript.

Response 1:

Thank you very much for your insightful comments concerning our manuscript. Those comments are very valuable and helpful for improving our paper, as well as our researches. We have studied all the comments carefully and have made corresponding corrections using the "Track Changes" function in the uploaded revised version.

As suggested, we have revised the Abstract and corrected the English expressions.

As for the SIMETAW model, we added more details and we explained the reason of choosing this model in our work (Please see the revised Materials and Methods Section). Also, added some discussion as suggested.

In the previous submitted manuscript version, the Table 3 was so large that we had thoughtlessly and unduly omitted the Slope value, which represents the tendency of historical changes in climatic variables. According to the Reviewer’s valuable suggestions, we added the slope of variation in climatic factors at 15 stations in Table 3, so as to make the analysis more clear and reliable.

In addition, we carefully checked other minor comments from the Reviewer shown in the manuscript. The corresponding revisions were shown in the revised manuscript using the "Track Changes" function, and the specific responses were listed as follows.

Point 2:

Some minor comments on English language errors shown in the manuscript:

Line 18: Climatic factors have obviously changing in different growth season of oat

Line 18: uptrend

Line 30: The impact of climate change on...

Line 30: is

Line 31: which should be payed attention for China and the rest of the word in the twenty-first century, especially, the Northeast and North China where has large areas of cultivated lands and irrigation areas, but has less than 20% of the total water resources.

Line 33~35: I would suggest "drought occurrence higher" replaces by '' frequently occurrence of drought" and " has caused" replaces by "is the cause of "

Line 59: Is it right? check it, please

Line 64: it should be "was"

Line 64: by explaining

Line 77: areas

Line 78: including

Line 86~87: It is difficult to understand, rewrite it, please

Line 98~99: unit

Line 117: a range of [number number]

Line 235: such as soil type,

Line 237: it should be plural

Line 244: as well as

>> For the above comments on English language errors, we have made corrections point-by-point in this revised version (Please see the "Track Changes" in the revised version; Lines 18, 30-32, 34, 77, 81-82, 96-97, 117-118, 134, 154, 281, 289).

Responses to the other comments concerning the logical and scientific issues:

Line22-24: Is this your results in this work? If not, rewrite it. You should say your results are useful and helpful for mitigating the adverse ... A hypothesis should not be your conclusion.

>> This statement seems to be not suitable as a conclusion, so we rewrote the last sentence of the Abstract. (L22-25 in the revised version)

Line37: Here should be cited

>> We added the cited reference: IPCC Technical Paper VI: Climate Change and Water. The IPCC Technical Paper indicates that, observed impacts of climate change on freshwater resources in parts of China, temperature increases and decreases in precipitation, along with increasing water use, have caused water shortages. (L36 and L365 in the revised version)

Line44-47: please, rewrite it.

>> We have rewritten this sentence and added corresponding references. (L64-66 in the revised version)

Line 47-54: Please, describing the details of the models and explaining the reason of choosing the model you used in this work.

>> Based on your suggestions, we added more details of the model and explained the reason of choosing the SIMETAW model in our work in the revised Materials and Methods Section (L116-124 in the revised version). Also, some necessary discussion were added. While in the Introduction Section, we just briefly introduced some commonly used models for evaluating crop water requirements.

Line 90~91: Is it right? please be carefully.

>> We have revised the inappropriate statements, and added more explicit descriptions for the functions of the SIMETAW model (L119-124 in the revised version), and some references were added.

Line 92~93: delete it, and also adding the references in each example.

>> We rewrote this sentence and added references. (L123-124 in the revised version)

Line 94~95: you should give more details of the model and all variables and parameters you used in this work

>> Thanks for your suggestions, we elaborated all the variables and parameters needed for model operation (L96-114 in Section 2.2 Data sources) and we explained the specific reason of choosing SIMETAW model in our study (in the Materials and Methods Section of the revised version).

Line 106~107: explaining this method

>> We explained this method and cited its references (L143-145 in the revised version)

Line 177~179: you have to show the data of climate variables in different years. It is impossible to see the increasing tendency or decline tendency using the data in table 3.

>> In the previous submitted manuscript version, the Table 3 was so large that we had thoughtlessly and unduly omitted the Slope value, which represents the tendency of historical changes in climatic variables. According to your nice suggestions, we added the slope of variation in climatic factors at 15 stations in Table 3 (Please see the revised Table 3 in Section 3.3)

Line 200~201: you should show the data of oat water requirement too.

>> The data of oat water requirement have actually been shown in Table 1. To avoid repetition, here we just added a reference to Table 1. (L234 in the revised version)

Line 231~232: you should compare the results from your work with other researchers' work.

>> We compared our work with other researchers’ work, and the results presented a relatively good consistency in terms of the key factors of climate impact on the water requirement. (L273-275 in the revised version)

Line 254: you should indicate P>0.01 or P>0.05.

>> We added P>0.05 in the text. (L299-300 in the revised version)

Line 262: could it be the start of a sentence? check it!

>> We rewrote the whole sentence to make it more logical. (L305-308 in the revised version)

Reviewer 2 Report

The paper covers the interesting issue of the assessment of water requirements of oat in a region of China under Climate Change. However the paper includes several drawbacks in taking logical conclusions. In few cases the Authors state the opposite of what they have stated few rows before. The overall conclusions seem rather surprising. The English needs a revision. 

Author Response

Point 1:

Comments and Suggestions for Authors

The paper covers the interesting issue of the assessment of water requirements of oat in a region of China under Climate Change. However the paper includes several drawbacks in taking logical conclusions. In few cases the Authors state the opposite of what they have stated few rows before. The overall conclusions seem rather surprising. The English needs a revision.

Response 1:

Thank you very much for your valuable comments on our manuscript. Those comments and suggestions are very helpful for improving the quality of our paper and research. We have studied all the comments carefully and made corrections using the "Track Changes" function in the uploaded revised version.

As the Reviewer suggested, we have modified many inconsistent statements in the previous manuscript, including some inappropriate words, sentences and expressions. We made major revisions to the Abstract, the Introduction, the Materials and Methods Section, Tables and related Result analysis, as well as the Discussion and Conclusions.

In addition, we corrected the English expressions errors based on the Reviewer’s suggestion, and we checked the English language in the whole text.

The main corrections and the responses to the Reviewer’s Comments are as follows:

Specific Comments Points & Our Responses:

R12: very bad English or several missing words

>> We revised the inappropriate expressions. (R12 in the revised version)

R32: correct the English

>> We modified this expression. (Please see the changes at R31-32 in the revised version)

R33: substitute semicolon with full stop.

>> It has been revised. (R33 in the revised version)

R48-49: GCMs are not meant nor used to “simulate the impacts of climate change on the water requirement of paddy rice, wheat and maize in various regions”. They (as in the name) are models that simulate the effect of climate drivers over land, oceans and atmosphere at the global (earth) level. Models as the ones mentioned in the following row are instead meant to do what the Authors wrote.

>> After checking the references, we found that as you mentioned above, GCMs are not used to simulate the impacts of climate change on the water requirement. Therefore, we revised the inappropriate expressions and rewrote the model related sentences, and their references were added. (R64-68 in the revised version)

R79: do harvest date, cultivated area and similar influence the “physical characteristics of oat”? I do not think so; in case, please specify how and mention some reference

>> Sorry for the misleading statements. The previous expression of “physical characteristics of oat” was not accurate, we mainly used these data to calculate the water requirement and irrigation demand of oat, but not the physical characteristics of oat. We have modified this sentence. (R113-114 in the revised version)

R87-88: correct the English

>> We revised the sentence (R118-119 in the revised version).

R87-88: the Authors give absolutely no information about the way and the assumptions the model SIMETAW has been used.

R87-88: the Authors state that they used the model SIMETAW; however the model developers state that the model uses “input information on crop and soil characteristics and the distribution uniformity of infiltrated irrigation applications in full or deficit conditions” (see below). The Authors of the present paper instead clearly confirm that they have no information about soil characteristics. Cite: Mancosu N., Spano D., et al. (2015), SIMETAW# - a Model for Agricultural Water Demand Planning, Water Resour. Manage., DOI 10.1007/s11269-015-1176-7.

>> Sorry for the misleading statements in the previous manuscript. In the Materials and Methods Section of this revised version, we added more details of the functions of SIMETAW model and we explained the reason of choosing the SIMETAW model in our work (R116-124 in the revised version). Also, some necessary references about the model were added.

>> As for the soil characteristics, we actually had collected and used the soil data when running the SIMETAW model (including the soil types, soil texture, wilting coefficient, soil bulk density, saturated moisture content, etc.). In the Section “2.2 Data sources” of this revised version, we added detailed information on the soil data for model operation. (R98-110 in the revised version)

R91: why? Explain or correct

>> The previous expression was not so convincing, so we rewritten the model-related statements. Please see the modified sentences between R119-124 in the revised version.

R92: cancel “reported”.

>> It has been modified. (R123-124 in the revised version)

R94: substitute “was” with “were”

>> It has been modified. (R126 in the revised version)

R98: usually ET is the worldwide adopted symbol for “EvapoTranspiration”, not daily rainfall

>> We replaced ET with Pcp, and we modified the Equation 3 (R133 in the revised version)

R108: it is not clear how the mentioned statistical techniques can be used to analyse the spatial variations. If sure about it, please clarify

>> Sorry for the confusing expressions. We have clarified the statements (R145-148 in the revised version). The spatial variations analysis were based on the IDW interpolation method using ARCGIS 10.0. SPSS statistics model was used to further analyze the correlations between oat water requirements and climatic factors.

R126: the mean annual value for Er (effective rainfall), even if not reported in the table1, is not higher in North China (calculated as 111.1 mm) but it is much higher in Northeast China, where it is 161.4 mm, then around 50% higher. ETaw how an opposite trend. Correct the text or the table

>> You are right. We have corrected the wrong analysis. (R163 in the revised version)

R160 (Figure 3) In box B the caption is CEr: it should be Er

>> Yes, we modified it in Figure 3, in this revised version.

R186: here Authors state the contrary of what stated at R19: which one I correct?

>> The statement here is correct. Compared to the other four climatic factors, the changes in mean precipitation and relative humidity of most sites are relatively not significant (P>0.05). We deleted the word “remarkable” and modified the sentence at R19 (R18 in the revised version).

R207: “Meanwhile” is rather misleading: it means “during the same time period”. The same in R63, R211, R228.

>> Thanks for your suggestion, we have made corrections. Please see the changes at R81, R250, R254, and R275 in the revised version.

R222: the word “thus” means that the statement that follows is a consequence of the previous statement. Here it is not the case.

>> We deleted it at the R268 in the revised version.

R224-225: it is correct to state that the decrease for mean wind speed and solar radiation leads to expect a decrease of water requirements, while an increase of mean temperature leads to the opposite direction; an increase of relative humidity is expected conversely to lower water requirement. The statement then is confusing and misleading.

>> Sorry for the confusing statements. Based on your suggestion, we have rewritten this sentence. Please see the changes at R269-272 in the revised version.

R230: substitute “geomorphic” with “geomorphologic”

>> It has been modified. (R277 in the revised version)

R231-233: these conclusions sound rather surprising: the irrigation requirement of oat is going to decrease and the Authors suggest to take actions to counteract this effect of Climate Change?

R259-262: incredible conclusions again (see R232)!!! Oat water requirement is expected to decrease: why would ever one desire to “counteract” such feature?

>> The decrease of crop water requirement can reduce the irrigation demand, but the effect of crop water requirement and irrigation on crop yield could be quite complex, which is not the main focus of this study. So we modified the inappropriate statements. Instead, we stated that: given the variability of different climatic factors, such as a significantly increased mean temperature during the growth season of oat, we need to take necessary strategies to adapt to the impacts of climate change on oat water requirements. Please see the revised expressions at R278-280, R304-308 in the revised version.

R254: here it is stated that relative humidity is decreasing; at R224 the Authors state the opposite.

>> Sorry for the misleading statements at R224. We have rewritten this sentence. Please see the changes at R269-272 in the revised version.

Round 2

Reviewer 1 Report

In my opinion, it can be accpted after minor revision following some advices:

Line 282-286 these are future work, not discussion.

More dicussions need to be descripted in a more comprehensive way, the main importance of the present study proposal, compared with the resuls of other reseachers.

Authors should indicate the value of coefficient in Equations, such as Kc, and add the calculation equations of the reference crop evapotranspiration and the daily water requirement of oat.

Author Response

Point 1:

Comments and Suggestions for Authors

In my opinion, it can be accepted after minor revision following some advices:

Line 282-286 these are future work, not discussion.

More discussions need to be descripted in a more comprehensive way, the main importance of the present study proposal, compared with the results of other researchers.

Authors should indicate the value of coefficient in Equations, such as Kc, and add the calculation equations of the reference crop evapotranspiration and the daily water requirement of oat.

Response 1:

Many thanks for your valuable comments on our manuscript again. We have studied all the comments carefully and made corrections using the "Track Changes" function in the uploaded revised version.

As the Reviewer suggested, we added detailed explanations on the value of Kc and CETc in Equations, and the calculation equation of ETo was added in the Materials and Methods Section. Besides, we made major revisions to the Discussion Section, in which we added a new paragraph to compare our results with other related researches.

Line 282-286 these are future work, not discussion.

>> We have deleted the previous statements of “Therefore, the influence of climatic factors on oat water requirement should be studied in more details in future research. The North China and Northeast China regions are relatively large, but the cultivation of oats does not span across the entire region………...” that are not relevant to the discussion.

More discussions need to be descripted in a more comprehensive way, the main importance of the present study proposal, compared with the results of other researchers.

>> We reorganized the Discussion Section, and particularly, we added a new paragraph to discuss our results with other related researches, and some necessary references were cited. (Please see the modified contents of the Discussion Section)

Authors should indicate the value of coefficient in Equations, such as Kc, and add the calculation equations of the reference crop evapotranspiration and the daily water requirement of oat.

>> In the Materials and Methods Section, we added the calculation equations or explanations of relevant parameters including ETo, CETc and Kc. Also, some necessary references were added. (Please see the supplemented contents between lines 101-117 in this revised version 
